# Association between Physical Function, Mental Function and Frailty in Community-Dwelling Older Adults: A Cross-Sectional Study

**DOI:** 10.3390/jcm13113207

**Published:** 2024-05-29

**Authors:** Hye-Jin Park, Ngeemasara Thapa, Seongryu Bae, Ja-Gyeong Yang, Jaewon Choi, Eun-Seon Noh, Hyuntae Park

**Affiliations:** Department of Healthcare and Science, Dong-A University, Busan 49315, Republic of Korea; hjpark3987@gmail.com (H.-J.P.); ngeemasara@gmail.com (N.T.); srbae@dau.ac.kr (S.B.); jgyang0702@gmail.com (J.-G.Y.); cjw4783@gmail.com (J.C.); dmstjssh3072@gmail.com (E.-S.N.)

**Keywords:** frailty, older adults, physical function, mental function, polypharmacy

## Abstract

**Background**: This study examines the relationship between physical and mental function and frailty, independently and in conjunction with polypharmacy, among older adults. **Methods**: This cross-sectional study consisted of 368 participants aged ≥60 years. The participants were categorized into either robust or frail groups using Fried’s frailty phenotype. Physical functions were assessed using grip strength, gait speed, Timed Up and Go (TUG), the Five Chair Sit to Stand Test (FCSST) and the Six-Minute Walk Test (SMWT). Mental functions were assessed using cognitive function and depression. Cognitive function was measured using Mini-Mental State Examination (MMSE). Depression was assessed with the Korean version of the Short Geriatric Depression Scale (SGDS). **Results**: The mean age of study population was 75.4 years. In this population, we identified 78.8% (*n* = 290) robust participants and 21.2% (*n* = 78) frail participants. The study examined frailty status (frail vs. non-frail) and frailty with and without polypharmacy using multivariate logistic regressions, adjusting for age and sex. In the logistic regression model estimating the risk of frailty, after adjustments for age, sex, BMI, and number of medications, individuals with low SMWT showed a significantly increased risk of frailty, with an odds ratio (OR) of 8.66 and a 95% confidence interval (CI) of 4.55–16.48. Additionally, global cognitive function was associated with a 1.97-fold increase in frailty risk (95% CI: 1.02–3.67). Moreover, in models adjusted for age, sex, and BMI to assess frailty risk linked to polypharmacy, the TUG, SMWT, and SGDS all showed increased risks, with ORs of 3.65 (95% CI: 1.07–12.47), 5.06 (95% CI: 1.40–18.32), and 5.71 (95% CI: 1.79–18.18), respectively. **Conclusions**: Physical function (SMWT, FCSST, TUG) and mental function (depression, cognition) were associated with frailty. By comprehensively examining these factors, we will gain valuable insights into frailty and enable more precise strategies for intervention and prevention.

## 1. Introduction

The global aging population is increasing steeply and is projected to double to 1.5 billion by 2036 [1,2]. The aging process is associated with a range of geriatric syndromes, and the coexistence of conditions such as frailty, cognitive impairment, and cardiovascular diseases (CVDs) exacerbates these syndromes. Frailty is a clinical condition characterized by a reduction in functional capacity, heightening an individual’s susceptibility to encountering adverse health outcomes when confronted with external stressors [3]. It is linked to a range of adverse consequences, encompassing cognitive deterioration, instances of falling, heightened apprehension about falls, hospitalization, increased medication usage, institutionalization, and mortality [4]. Therefore, early recognition and prevention of frailty is important.

In clinical settings, several frailty screening tools are utilized, including Fried’s Frailty Phenotype [5], also referred to as the Cardiovascular Health Study (CHS) index [6], the FRAIL scale [7], the Frailty Index of Accumulative Deficits (FI-CD) [8], and the Clinical Frailty Scale. In older populations, based on the screening tools, systematic reviews have reported the prevalence of prefrailty ranging from 35% to 50% and frailty from 7% to 12%, which increases to 26% among those over 85 years old [9,10]. Fried’s Frailty Assessment Tool [5] is the most commonly and widely used for frailty screening, and includes five physiological measures: diminished walking speed, reduced physical activity, unintentional weight loss, low energy levels, and weak grip strength [5]. The presence of 1–2 of these criteria suggests a prefrail state, and the presence of 3 or more of the 5 criteria indicates frailty.

Prefrail/frail individuals have been observed to exhibit significantly poorer performance in upper limb dexterity, lower limb power, balance, and endurance [11]. However, the current frailty criteria only incorporate grip strength and gait speed as markers of physical performance. Furthermore, a substantial decline in aerobic capacity can be noted after the age of 40, with up to a 30% reduction by the age of 65 in older adults [12]. Despite the decline in aerobic capacity occurring with each passing decade, physical exhaustion tends to manifest much later in the frailty cycle [13,14]. The existing frailty criteria, which assess self-reported exhaustion, may not be sufficiently accurate in predicting early signs of decline in aerobic capacity. Such deterioration in physical fitness contributes to functional dependence [15].

Frailty is a dynamic state that impacts not only physical domains but also psychological and social domains [16,17]. Depression, one of the prevalent psychological disorders among older adults, exhibits a strong association with frailty [18]. Frail individuals are more prone to depression due to their diminished physical function, reduced social engagement, and lower levels of social support [19,20]. Moreover, physical frailty may be associated not only with depression but also with cognitive impairment. Several studies have indicated a heightened risk of cognitive decline among individuals with depression. Furthermore, epidemiological studies and clinical investigations have highlighted frailty as a significant risk factor for cognitive impairment [21,22].

Therefore, the primary objective of our study was to explore and comprehensively investigate the correlation between frailty and cognitive and mental function, as well as to identify the connection between frailty and various physical fitness metrics. Notably, frailty is also associated with multiple negative health outcomes [3] which might require medications, thereby increasing the likelihood of medication-related side effects. Consequently, frail individuals are more likely to develop polypharmacy. A longitudinal study revealed that participants taking four or more medications faced a 55% elevated risk of frailty [23]. Moreover, polypharmacy could potentially accelerate the onset of frailty by negatively impacting factors included in the definition of frailty, including muscle loss [24], cognitive function [25], mental health [26], and various other comorbidities. Therefore, this study also investigated the relationship between physical fitness measures, and the risk of developing frailty alongside polypharmacy.

## 2. Materials and Methods

### 2.1. Study Population

This multi-center cross-sectional study screened 488 participants aged 60 years and older. A total of 368 individuals met the recruitment criteria. We included all patients aged ≥60 years who can perform independent activities of daily living who visited the Digital Healthcare Lab at Dong-A University in March 2022 to March 2023. The exclusion criteria applied are explained in Figure 1. A total of 120 participants were excluded following those criteria. The participants were required to sign written informed consent at the beginning of the study. Ethical approval for this study was granted by the Institutional Review Board of Dong-A University on 24 March 2022 (IRB No. 2-1040709-AB-N-01-202201-HR-008-02).

### 2.2. Outcome Measure and Predictor Variables

#### 2.2.1. Frailty

Frailty was defined based on Linda Fried’s frailty phenotype guidelines [5]. The same method was used in our previous study [27]. Fried’s phenotype consists of five conditions as follows:i.Weakness—measured by grip strength, adjusted for sex and body mass index (BMI);ii.Weight loss—unintentional weight loss of 4.5 kg in the previous year;iii.Slowness of gait—calculated by the time taken to walk 4 m twice without assistance;iv.Exhaustion—self-reported exhaustion;v.Reduced physical activity—calculated based on accelerometer data as per the recommendation for physical activity in older adults by the American College of Sports Medicine.

The frailty risk group was defined according to Linda Fried’s criteria as including both the frail and prefrail groups. In other words, participants who met any one of the previously mentioned criteria were categorized into the frailty risk group, while those who did not meet any criteria were considered part of the robust group.

#### 2.2.2. Frailty with Polypharmacy

Polypharmacy was defined as an intake of >4 medications [23]. The medication intake number was recorded from the participant’s prescriptions. Medication used on a regular basis, excluding vitamins and mineral supplements, was recorded. The participants were further categorized into four groups based on presence and absence of frailty, polypharmacy, and/or both, as follows:i.Frail with presence of polypharmacy (FP);ii.Frail and absence of polypharmacy (FNP);iii.Robust with presence of polypharmacy (NFP);iv.Robust and absence of polypharmacy (NFNP).

#### 2.2.3. Physical Function

The following variables were used as a measure of physical function:i.Muscle strength: A grip strength test was used to analyze muscle strength with a digital hand dynamometer (TKK 51Grip-D, Takei, Tokyo, Japan). Participants were directed to keep their shoulders slightly distanced from their body while ensuring that the dynamometer was oriented downward during the test. The test was conducted twice, alternating between the right and left hand each time. The participants were motivated to exert their optimal effort during the test to achieve the most favorable outcome. The highest recorded value then represented the individual’s maximum handgrip strength.ii.Gait speed: Gait speed was assessed using a 4 m walk where the participants were instructed to walk at their normal speed. The use of an assistive aid was allowed if habitually used. This test included a 1.5 m acceleration phase, 4 m walk, and a 1.5 m deceleration phase. The timing was only applied to 4 m walk.iii.Functional mobility: A Timed Up and Go (TUG) test was used to assess functional mobility. This test records the time it takes a person to rise from a chair, walk three meters, perform a turn, and then return to a seated position. Before the commencement of the test, participants were seated on a chair. Upon a signal, participants performed the test where they were instructed to walk at a brisk pace but not to run. Multivariate analysis was performed for each group by dividing the data into two groups, low function (≥7.20 s) and high function (<7.20 s).iv.Lower limb strength: A Five Chair Sit to Stand Test (FCSST) was used to assess lower limb strength. During the FCSST, participants were instructed to rise from a chair and sit down consecutively five times as quickly as possible with their arms crossed over their chest. The speed at which they completed this test was timed. Multivariate analysis was performed for each group by dividing the data into two groups, low function (≥9.35 s) and high function (<9.35 s).v.Endurance capacity: A Six-Minute Walk Test (SMWT) was used to assess endurance capacity. In this assessment, participants were asked to walk at a steady pace for 6 min with the goal of covering as much distance as possible within this time frame. Multivariate analysis was performed for each group by dividing the data into two groups, low function (≤380 m) and high function (>380 m).

#### 2.2.4. Mental Function

The Mini-Mental State Examination (MMSE) was used as a measure of global cognitive function. The MMSE has a total score of 30, and the test consists of time orientation (5), spatial orientation (5), memory recall (3), language (16), and space-time configuration (1). Lower scores correspond to poorer cognitive function and were dichotomized and multivariate analyzed based on a previous study [28].

Depression was assessed with the Short Geriatric Depression Scale (SGDS), which is a 15-question self-report depression scale used to diagnose depression [29,30,31]. The total score ranges from 0 to 15, with higher scores reflecting greater depressive symptoms. The scores were dichotomized and multivariate analyzed based on a previous study [32].

#### 2.2.5. Short Physical Performance Battery (SPPB)

The Short Physical Performance Battery (SPPB) is a comprehensive assessment tool that measures three key physical capabilities: static balance, gait speed, and the ability to rise from a chair. Each of these abilities is quantified through timed measurements. For the static balance assessment, participants were required to assume three progressively challenging postures for 10 s each: (i) side-by-side, (ii) semi tandem (heel of one foot adjacent to the big toe of the other), and (iii) tandem (heel of one foot directly in front of and touching the other foot). The gait speed was assessed twice by measuring the time taken by participants to walk a 4 m distance at their usual pace. The fastest of two attempts was recorded for the final score calculation. In the chair rise test, participants were instructed to demonstrate their ability to stand from a seated position as quickly as possible, with their arms folded across their chest. The time taken to complete this action was recorded.

#### 2.2.6. Resting Blood Pressure (BP)

Resting BP was assessed at the baseline using oscillometer techniques, with a BP device (BPBIO320S, InBody Co., Seoul, Republic of Korea). The measurements were taken with a blood pressure cuff properly applied to the subject’s dominant arm in accordance with current clinical guidelines [33]. BP was measured twice after a 30 min rest period in a seated position. The mean of these measurements was calculated and recorded [34].

### 2.3. Other Variables

Anthropometric measurements—height, weight, BMI, and socio-demographic measures—age, sex, education, and living status were also measured.

### 2.4. Statistical Analyses

Statistical analyses were performed using IBM SPSS V28.0 (IBM Corp., Armonk, NY, USA). For continuous variables with normal distributions, Student’s *t*-tests were used to assess differences between groups. Conversely, the Mann–Whitney U test was employed for continuous variables that were not normally distributed. Categorical variables were analyzed using the Chi-square test. For the normally distributed data, we reported means ± standard deviations and medians (interquartile ranges) for non-normally distributed data. Categorical variables were quantified as frequencies and percentages. Differences in physical, cognitive, and mental functions across groups were assessed using one-way Analysis of Covariance (ANCOVA) for normally distributed variables, and the Kruskal–Wallis test for non-distributed variables. Multivariate logistic regression analysis was used to calculate odds ratios (ORs) with 95% confidence intervals (CIs) to assess the combined effect of frailty and polypharmacy on physical and mental function. All significance levels were set to *p* < 0.05.

## 3. Results

The demographic, anthropometric, physical, and mental characteristics of the participants are summarized in Table 1. There were 290 robust participants (78.8%) and 97 frail participants (21.2%). The frail participants had significantly higher anthropometric measures (BMI) and age than the robust participants. The frail participants were also observed to have significantly decreased physical function (grip strength, gait speed, TUG, SMWT, FCSST) and global cognition (MMSE) compared to robust. In addition, depressive symptoms identified by SGDS were higher in frail participants, and the number of medications was higher in frail participants than in robust participants.

Figure 2 shows that TUG, SGDS, and SMWT exhibit significant differences between the groups (FP, OFNP, NFP, and NFNP), regardless of polypharmacy status. In contrast, FCSST and MMSE scores appear to vary depending on the polypharmacy status.

Table 2 presents the results of the logistic regression analysis investigating risk factors for frailty. SMWT was associated with the greatest increase in frailty risk (OR: 7.17, 95% CI: 6.59–22.32). Similarly, individuals with poor performance on TUG (OR: 2.70, 95%CI: 1.41–5.19) and FCSST (OR: 2.36, 95% CI: 1.22–4.58) were also at significantly increased risk of frailty. Low global cognitive function (OR: 2.02, 95% CI: 1.06–3.83) and high depressive symptoms (OR: 2.10, 95% CI: 1.05–4.22) were also significant risk factors. After adjusting for age, sex, and the number of medications taken, the risk of frailty increased for SMWT (OR: 8.66, 95% CI: 4.55–16.48) and low cognitive function (OR: 1.97, 95% CI: 1.02–3.67).

Table 3 shows results from the logistic regression model estimating the risk of frailty with polypharmacy. Individuals with low SMWT had the strongest association with the risk of frailty with polypharmacy (OR: 6.41, 95% CI: 1.98–20.72). High SGDS also showed a significantly high risk of frailty with polypharmacy (OR: 5.29, 95% CI: 1.83–15.26). After adjusting for age, sex, and BMI, the risk of frailty with polypharmacy remained significant for those with low SMWT (OR: 5.06, 95% CI: 1.40–18.32), poor TUG (OR: 3.65, 95% CI: 1.07–12.47), and high SGDS (OR: 5.71, 95% CI: 1.79–18.18).

## 4. Discussion

This cross-sectional study investigated the associations between physical function (including muscle strength, gait speed, functional mobility, lower limb strength, and endurance capacity), cognitive function, and depression in older adults, and the estimated risk of frailty. We also examined whether these factors, along with polypharmacy medication use, have a combined influence on frailty risk.

In our study, the frailty risk group performed poorly on physical function tests (SMWT, TUG, and FCSST) and mental function tests (MMSE and SGDS) compared to the robust group. We also observed lower endurance capacity in the frailty risk group compared to the robust group. A previous study has shown that frail individuals exhibit increased energy expenditure per meter of walking distance, highlighting an important measure of energy demand [35]. They also make greater use of muscle groups to compensate for gait deficits due to muscle weakness and balance problems, which are recognized as inefficient and costing energy [36]. The SMWT is a measure of functional ability and endurance capacity, and the frailty scores highly correlate with it [37]. While endurance capacity declines with each passing decade, physical exhaustion tends to appear much later in the frailty cycle [13,14]. This makes existing frailty criteria that assess self-reported exhaustion inadequate for predicting early signs of decline in endurance capacity.

The TUG test is a collective assessment of strength in lower extremities, gait speed, balance, and cognition [38], that captures a variety of age-related physiological changes and differentiates frail older adults from non-frail older adults [39]. Previous studies on community-dwelling older adults have shown that frail older adults take longer to complete the TUG than non-frail older adults [40,41]. TUG is particularly sensitive in predicting frailty when it takes more than 8 s to complete [42,43]. Our results (≥7.20 s) indicate that frailty might be identifiable in our population at a slightly lower threshold. The FCSST is an important marker of physical independence and lower extremity physical function [44,45]. Previous studies have found that the time taken to complete the FCSST is a stronger predictor of functional impairment and disability [46,47] and falls [48,49].

Beyond the physical aspect, cognitive function and depression have been established to be equally important aspects of frailty. The frailty risk group had lower global cognitive function compared to the robust group. Although this study was not able to identify a causal relationship, cognitive impairment can lead to limitations in activities of daily living, reduced participation in physical and social activities, potentially contributing to physical frailty [50]. In addition, individuals with lower cognitive function and depression were found to have a 1.97- and 1.90-times higher risk of frailty, respectively, consistent with the previous findings [51,52]. Depressive symptoms were also found to be more prevalent in the frailty risk group, further highlighting the multifaceted nature of frailty. Depression can negatively affect an individual’s motivation for physical activity and self-care, thereby worsening physical function and increasing frailty risk [39]. Particularly, SMWT (8.66-fold) and MMSE (1.97-fold) showed even stronger associations with frailty risk after adjusting for age and sex.

The average medication consumption of frail patients is higher than that of healthy patients [53,54,55]. Therefore, we further investigated the association between physical function, cognition, and depression and the risk of frailty with polypharmacy. Meta-analytic studies have shown that polypharmacy increases depression risk [26] and depression is linked to frailty [51]. The results of this study indicate that individuals with depression have a 5.71-fold higher risk of frailty associated with polypharmacy compared to those without depression. These findings support previous research suggesting that depression has a greater impact on the risk of frailty in individuals with polypharmacy compared to those with frailty alone.

The SMWT results are also influenced by several factors, including the number of chronic diseases [56]. Previous studies have shown a linear relationship between the medication intake number and balance impairment, with the time taken to complete the TUG test increasing with the number of medications taken [57]. This association is further supported by our finding that a slower TUG performance, which assesses lower extremity strength, gait speed, and balance [39], was associated with a 3.65-fold increased risk of frailty in individuals taking multiple medications.

Physical frailty is primarily characterized by characteristics such as decreased muscle strength, endurance capacity, and the lower extremity mobility. The functional decline significantly impacts daily activities, leading to increased dependency and a decreased quality of life for the individual. Linda Fried’s frailty phenotype does not fully describe the physical and mental functions associated with frailty identified in this study. In this study, the SMWT, TUG, and FCSST were found to have significant relationships in predicting frailty, suggesting that these tests are effective tools for assessing functional decline in frail individuals.

Furthermore, we established thresholds for the SMWT, FCSST, and TUG tests to identify physical function limitations in frail community-dwelling older adults. The SMWT with a cut-off score of ≤380 m estimates endurance capacity and is proposed as an independent predictor of frailty risk. Similarly, a slow TUG (≥7.20 s) and slower FCSST (≥9.35 s), which are important indicators of physical independence, are proposed as independent risk factors for frailty. Furthermore, it is proposed that cognitive decline and depression, in addition to physical function, are predictors of frailty risk. Assessing the state of decline in physical function and cognitive and mental function is crucial for implementing interventions aimed at delaying functional decline or reversing frailty. The proposed screening tool will facilitate identification, allowing for tailored interventions to enhance physical, cognitive, and mental function, ultimately reducing the burden of frailty in the community.

Although this study provides valuable insights to our understanding of the relationship between frailty, polypharmacy, and health outcomes in older adults, there are limitations to be acknowledged. Firstly, the cross-sectional design of our study limits the ability to establish causality. Longitudinal studies are needed to clarify the temporal dynamics between frailty, polypharmacy, mental and functional decline. Additionally, the participant pool was limited to a specific demographic group, which may limit the generalizability of the findings. The study did not assess specific drug classes or possible drug interactions that may have different effects on frailty and functional outcomes. We also did not consider the specific comorbidities of chronic diseases that may affect endurance. Future research should include studies that examine the impact of specific medications and polypharmacy patterns on frailty and its interaction with frailty risk factors in older adults, as well as RCTs that improve upon these findings.

## 5. Conclusions

This study highlights the association between frailty risk and both physical and mental health risk factors. It demonstrates, specifically, that cardiopulmonary endurance and lower limb performance, as measured by SMWT, FCSST, and TUG, along with mental health assessed by SGDS and the MMSE, are significantly linked to frailty. Furthermore, the findings indicate that lower cardiopulmonary endurance, lower limb function, and higher depressive scores present a higher risk of frailty, particularly in individuals with polypharmacy. This research underscores the importance of a comprehensive assessment of both physical and mental health risk factors to facilitate more accurate decisions in the community-based prevention and management of frailty.

## Figures and Tables

**Figure 1 jcm-13-03207-f001:**
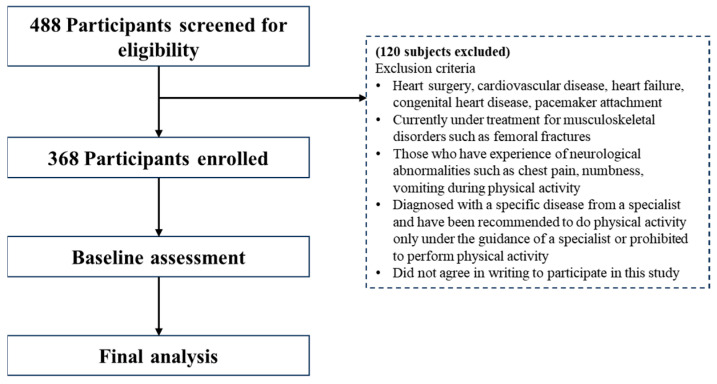
Flowchart of screening and enrollment of study participants.

**Figure 2 jcm-13-03207-f002:**
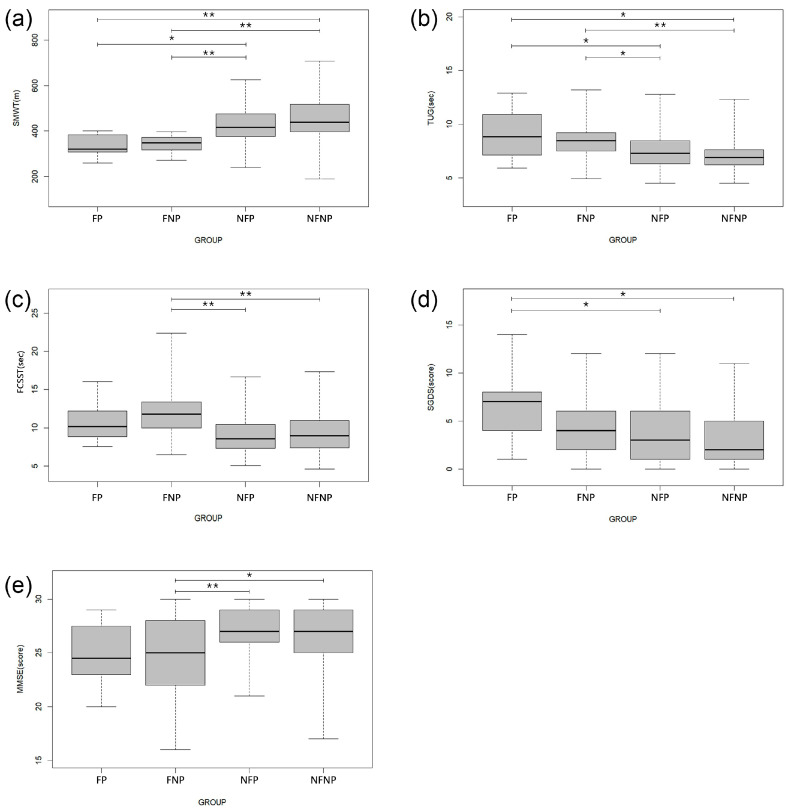
Significant differences in frailty status and/or polypharmacy according to physical, mental, and cognitive function; (**a**) Six-Minute Walk Test, (**b**) Timed Up and Go, (**c**) Five Chair Sit to Stand Test, (**d**) Short Form of Geriatric Depression Scale, (**e**) Mini-Mental State Examination. One-way analysis of covariance (ANCOVA) with the Bonferroni test and a Kruskal–Wallis test with the Mann–Whitney U test were used compare frail with presence of polypharmacy (FP), frail and absence of polypharmacy (FNP), robust with presence of polypharmacy (NFP), and robust and absence of polypharmacy (NFNP). The error bars represent the standard error of the mean. * *p* < 0.05, ** *p* < 0.001.

**Table 1 jcm-13-03207-t001:** Demographic, anthropometric, physical, and mental characteristics of all subjects.

Variables	Total (*n* = 368)	
	Robust (*n* = 290)	Frail (*n* = 78)
Age (year)	74.5 ± 5.4	78.6 ± 4.4 *
Male/Female (*n*)	77/213	20/58
Height (m)	1.6 ± 0.1	1.5 ± 0.1 *
Weight (kg)	59.9 ± 9.1	61.3 ± 8.5
Body Mass Index (kg/m^2^)	24.2 ± 3.1	25.4 ± 3.0 *
Systolic Blood Pressure (mmHg)	134.7 ± 16.9	138.8 ± 19.8
Diastolic Blood Pressure (mmHg)	73.4 ± 11.7	74.5 ± 12.6
Education (year)	10 ± 2.1	12 ± 3.0
Living alone ^a^ *n* (%)	226 (78)	69 (89)
Grip strength (kg)	25.1 ± 6.2	21.2 ± 6.4 *
Gait speed (m/s)	1.3 ± 0.3	1.0 ± 0.1 *
Timed Up and Go (s)	7.1 ± 1.3	8.6 ± 1.9 *
Six Minute Walk Test (m)	449.4 ± 94.9	346.4 ± 49.9 *
Five Chair Sit to Stand Test (s)	9.2 ± 2.7	11.7 ± 3.1 *
Short Physical Performance Battery (score)	11. 2 ± 1.1	10.2 ± 1.3 *
Polypharmacy (%)	6.7	16.4
Short Form of Geriatric Depression Scale ^b^ (score)	3.5 (7.5)	5.0 (9.5) *
Mini-Mental State Examination (score)	26.5 ± 2.7	24.8 ± 3.4 *

Data are represented as mean ± standard deviation, median (interquartile), and *n* (%). Variables were compared by *t*-test, ^a^ Chi-square test, ^b^ Mann–Whitney test, * *p* < 0.05.

**Table 2 jcm-13-03207-t002:** Adjusted odds ratio (95% confidence interval) for the estimated risk of frailty based on physical and mental function divided by the cut-point.

Variables		Unadjusted		Adjusted	
	*n*	OR (95%CI)	*p*-Value	OR (95%CI)	*p*-Value
TUG (s)					
High	188				
Low (≥7.20)	180	2.70 (1.41–5.19)	0.031	1.87 (0.93–3.78)	0.137
SMWT (m)					
High	254				
Low (≤380)	114	7.17 (6.59–22.32)	0.007	8.66 (4.55–16.48)	0.001
FCSST (s)					
High	185				
Low (≥9.35)	183	2.36 (1.22–4.58)	0.027	1.81 (0.89–3.67)	0.096
MMSE (score)					
High	271				
Low (≤24)	97	2.02 (1.06–3.83)	0.025	1.97 (1.02–3.67)	0.016
SGDS (score)					
Low	312				
High (≥9)	56	2.10 (1.05–4.22)	0.020	1.70 (0.81–3.59)	0.224

Odds ratio (95% confidence interval) adjusted for age, sex, BMI and number of medication intake; TUG, Timed Up and Go; SMWT, Six-Minute Walk Test; FCSST, Five Chair Sit to Stand Test; SGDS, Short Form of Geriatric Depression Scale; MMSE, Mini-Mental State Examination; OR: Odds Ratio, CI: Confidence Interval.

**Table 3 jcm-13-03207-t003:** Adjusted odds ratio (95% confidence interval) for the estimated risk of frailty with polypharmacy based on physical and mental function divided by the cut-point.

Variables		Unadjusted		Adjusted	
	*n*	OR (95%CI)	*p*-Value	OR (95%CI)	*p*-Value
TUG (s)					
High	188				
Low (≥7.20)	180	2.89 (0.88–9.49)	0.079	3.65 (1.07–12.47)	0.039
SMWT (m)					
High	254				
Low (≤380)	114	6.41 (1.98–20.72)	0.002	5.06 (1.40–18.32)	0.013
FCSST (s)					
High	185				
Low (≥9.35)	183	0.14 (0.13–1.33)	0.138	0.35 (0.10–1.18)	0.091
MMSE (score)					
High	271				
Low (≤24)	97	1.87 (0.63–5.62)	0.262	1.78 (0.57–5.23)	0.320
SGDS (score)					
Low	312				
High (≥9)	56	5.29 (1.83–15.26)	0.002	5.71 (1.79–18.18)	0.003

Odds ratio (95% confidence interval) adjusted for age, sex, BMI; TUG, Timed Up and Go; SMWT, Six-Minute Walk Test; FCSST, Five Chair Sit to Stand Test; SGDS, Short Form of Geriatric Depression Scale; MMSE, Mini-Mental State Examination; OR: Odds Ratio, CI: Confidence Interval.

## Data Availability

Qualified researchers can obtain the data from the corresponding author (htpark@dau.ac.kr). The data are not publicly available due to privacy concerns imposed by the IRB.

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
