# Peer review of "Association between Physical Function, Mental Function and Frailty in Community-Dwelling Older Adults: A Cross-Sectional Study"

_jcm, 2024, doi:10.3390/jcm13113207_

Round 1

Reviewer 1 Report

Comments and Suggestions for Authors

Journal of Clinical Medicine jcm-2962564

Title: Association between Physical Function, Mental Function and Frailty in Community-dwelling Older Adults: A Cross-sectional Study

This cross-sectional study describes the association between frailty and various physical metrics, and cognitive and mental function, besides those impacts on frailty with polypharmacy. Frailty is an important geriatric syndrome that affects the prognosis of older people, and it is important to evaluate its multifaceted components. However, I think this paper has many problems to be improved as indicated below.

1.     Introduction

The introduction part should include the background of the study, the known facts, the facts not yet fully known, and the purpose of this study, in that order. This manuscript describes the known facts well but does not mention what is not fully assessed. In addition to that, the authors discuss opinions that are not based on the results of the study (as L61-62), which should be discussed in the Discussion part, not in the Introduction.

2.     Line 87

Please indicate whether it is a multicenter or single-center study.

3.     Line 88

There is some confusion in the description of the entire population, inclusion criteria, and exclusion criteria. The reviewer suggests writing as follows. “We included all patients aged 60 years or older with independent ADLs who visited XX medical center in MM, YYYY to MM, YYYY. The inclusion criteria are ~, and the exclusion criteria are ~.”

4.     Line 97 Figure1

 Was the assessment of physical function done solely by self-report? Please describe whether any examinations or tests were performed on physical endurance before measuring the following metrics.

5.     Line 114-116

  The authors have assessed frailty according to Fried’s phenotype and divided the patients into robust and frail groups, but did you not consider a pre-frail group? Why not also separate the pre-frail group and analyze the results of physical function measurement?

6.     L133-139

  Please describe the thresholds for the low-function and high-function groups.

7.     L141-146

  I suppose the authors measured the time to finish FCSST, so please specify. The authors should also describe the thresholds for the low-function and high-function groups.

8.     L148-152

  Please describe the thresholds for the low-function and high-function groups.

9.     Line174

Referring to Table 1, it seems that blood pressure measurement and SPPB are also performed. Please clearly state these measurements.

10.  Line 198 

  Medication dosage is not described in Table 1. If the authors measured that, it should be clearly stated in the Materials and Methods part and the Result part.

11.  Line 202 Table1

  Mean ± standard deviation can be used only when the variable is normally distributed; in the case of a non-normal distribution, it should be the median (interquartile range).

12.  Line 205

One-way ANCOVA assumes a normal distribution of the population; SGDS was probably non-normally distributed, so please check carefully that your statistical methods are correct and modify them if necessary.

13.  Line 207-211

  The results and their interpretation are confusing. TUG, SGDS, and SMWT show differences between the groups based on the presence or absence of frailty, regardless of polypharmacy. On the other hand, for FCSST and MMSE, the effect of frailty differs between those with polypharmacy and those without polypharmacy, which indicates the combined relationship of frailty and polypharmacy.

14.  Line 221

  Please correct the two periods.

15.  Line 226-228

  According to Table 3, After adjustment, the odds ratio might be 8.66 for a shorter 6-minute walk distance and 1.97 for a lower MMSE score. Please correct the data.

16.  Line248

Is this sentence grammatical? How about writing “In our study, we found~” or “Our study showed~”.

17.  Line259-261

The authors repeat the same content in two sentences, so either one is fine.

18.  Line270-294

In the Discussion part, the authors’ logic is confusing and should be thoroughly reviewed. I believe the purpose and conclusion of this study is to address the issue that the current diagnostic criteria for frailty do not take into account the loss of endurance, so please discuss how to draw that conclusion. Currently, you are merely listing past studies.

The first result of this study is that lower extremity endurance was strongly associated with frailty and also with a combination of frailty and polypharmacy; the second is that there was also an association between mental and cognitive function and physical function. Furthermore, in the group in which polypharmacy was combined with frailty, the effect of SGDS was different from that of frailty alone. For each of these, you should describe what actions can be taken if early recognition is made, citing proper previous studies.

19.  Line 295-303

Wouldn't it be a LIMITATION to not take into account chronic diseases that do not affect exercise endurance?

Lastly, while the findings presented in this study should be of interest to JCM’s audience, the reviewer thinks that a substantial revision, based on the above comments, is needed to make this manuscript suitable.

Author Response

Response to Reviewer’s comments / Manuscript ID: jcm-2962564

Reviewer #1

Q1. [L61-62] The introduction part should include the background of the study, the known facts, the facts not yet fully known, and the purpose of this study, in that order. This manuscript describes the known facts well but does not mention what is not fully assessed. In addition to that, the authors discuss opinions that are not based on the results of the study (as L61-62), which should be discussed in the Discussion part, not in the Introduction.

Response1: Thank you for your comments. I have removed and revised the sections you mentioned from the introduction, and I have added the unnecessary content from the introduction to the discussion as per your suggestions.

Q2. [87] Please indicate whether it is a multicenter or single-center study.

Response2: Thank you for your comment. We have revised:

Line [108] This multi-center cross-sectional study screened 488 participants aged 60 years and older. 368 individuals met the recruitment criteria.

Q3. [88] There is some confusion in the description of the entire population, inclusion criteria, and exclusion criteria. The reviewer suggests writing as follows. “We included all patients aged 60 years or older with independent ADLs who visited XX medical center in MM, YYYY to MM, YYYY. The inclusion criteria are ~, and the exclusion criteria are ~.”

Response 3: Thank you for your comment. As per your comment we revised the sentence as follows: Line [108-115]

This multi-center cross-sectional study screened 488 participants aged 60 years and older. 368 individuals met the recruitment criteria. We included all patients aged 60 years or older with independent ADLs who visited Digital Healthcare Lab at Dong-A University in 03. 2022 to 03. 2023. The exclusion criteria applied are explained in Figure 1. A total of 120 participants were excluded following those criteria. All participants provided signed informed consent at the beginning of the study. Ethical approval for this study was granted by the Institutional Review Board of Dong-A University on March 24, 2022 (IRB No. 2-1040709-AB-N-01-202201-HR-008-02).

Q4. [91, figure1] Was the assessment of physical function done solely by self-report? Please describe whether any examinations or tests were performed on physical endurance before measuring the following metrics.

Response 4:  Thank you for your comments.  All physical functions were measured in accordance with the methods section.: Line [175-213]

2.2.3. Physical Function

The following variables were used as a measure of physical functions:

  1. Muscle strength: Grip strength test was used to analyze muscle strength using a digital hand dynamometer (TKK 51Grip-D, Takei, Tokyo, Japan). During the test, participants were directed to keep their shoulders slightly distanced from their body while ensuring that the dynamometer was oriented downward. The test was conducted twice, alternating between the right and left hand each time. The participants were motivated to exert their optimal effort during the test to achieve the most favorable outcome. The highest recorded value then represents the individual's maximum handgrip strength.
  2. Gait speed: Gait speed was assessed using a 4-m walk where the participants were instructed to walk at their normal speed and were allowed to use a walking aid if they normally used one. This test included a 1.5-m acceleration phase, 4-m walk, followed by a 1.5-m deceleration phase. The timing was only applied to 4-m walk.

iii. Functional mobility: Timed Up and Go (TUG) test was used to assess functional mobility. This assessment involves recording the time it takes a person to rise from a chair, walk three meters, perform a turn, and then return to a seated position. Before the commencement of the test, participants were seated on a chair before. When a signal was given, participants performed the test where they were asked to walk at a brisk pace without running. Multivariate analysis was performed for each group by dividing the data into two groups, low function (≥7.20), and high function.

  1. Lower limb strength: Five Chair Sit to Stand Test (FCSST) as used to assess lower limb strength. During FCSST, participants were instructed to stand up from a chair as quickly as possible and sit down consecutively five times and the speed at which they completed this test was timed. The participants were required to cross their arms in front of their chest while performing this test. Multivariate analysis was performed for each group by dividing the data into two groups, low function (≥9.35), and high function.
  2. Endurance capacity: Six-minute walk test (SMWT) was used to assess endurance capacity. In this assessment, participants are asked to walk at a steady pace for 6 minutes with the goal of covering as much distance as possible within this time frame. Multivariate analysis was performed for each group by dividing the data into two groups, low function (≤380), and high function.

Q5. [114-116] The authors have assessed frailty according to Fried’s phenotype and divided the patients into robust and frail groups, but did you not consider a pre-frail group? Why not also separate the pre-frail group and analyze the results of physical function measurement?

Response 5: Thank you for your insightful comments. In this study, the prevalence of frailty among the community-dwelling elderly population was relatively low, approximately 4%, which is consistent with the findings of a meta-analysis indicating a frailty prevalence of 3.9% in AISA. Previous research has often analyzed frailty by grouping pre-frail and frail individuals together to assess the broader spectrum of frailty risk. Consequently, to maintain consistency with these methodologies and due to the low prevalence rate, we combined the pre-frail and frail statuses into a single category to better identify those at risk of frailty. This approach aligns with established research practices and enhances the robustness of our findings by focusing on a unified risk group.

[Line145-148] The frailty risk group was defined according to Linda Fried's criteria as including both the frail and prefrail groups. In other words, participants who met any one of the previously mentioned criteria were categorized into the frailty risk group, while those who did not meet any criteria were considered part of the robust group.

References include:

1.Sewo Sampaio, P.Y., et al., Differences in lifestyle, physical performance and quality of life between frail and robust Brazilian community‐dwelling elderly women. Geriatrics & gerontology international, 2016. 16(7): p. 829-835.

2.Bennett, A., et al., Prevalence and impact of fall-risk-increasing drugs, polypharmacy, and drug–drug interactions in robust versus frail hospitalised falls patients: a prospective cohort study. Drugs & aging, 2014. 31:

3.Siriwardhana, D. D., Hardoon, S., Rait, G., Weerasinghe, M. C., & Walters, K. R. (2018). Prevalence of frailty and prefrailty among community-dwelling older adults in low-income and middle-income countries: a systematic review and meta-analysis. BMJ open, 8(3), e018195.

Q6. [133-139] Please describe the thresholds for the low-function and high-function groups.

Response 6: Thank you for the comment. We added the describe the thresholds.  Line [199-200]

Multivariate analysis was performed for each group by dividing the data into two groups, low function (≥7.20 sec), and high function (7.20 sec).

Q7. [141-146] I suppose the authors measured the time to finish FCSST, so please specify. The authors should also describe the thresholds for the low-function and high-function groups.

Response 7: Thank you for the comment. We added the describe the thresholds.  Line [207-208]

Multivariate analysis was carried out by categorizing the data into two groups: low function (≥9.35) and high function (<9.35) for each group.

Q8. [148-152] Please describe the thresholds for the low-function and high-function groups.

Response 8: Thank you for the comment. We added the describe the thresholds.

Line [214] Multivariate analysis was conducted for each group by separating the data into two categories: low function (≤380m) and high function (>380m).

Q9. [174] Referring to Table 1, it seems that blood pressure measurement and SPPB are also performed. Please clearly state these measurements.

Response 9: Thank you for the comment. We added the information regarding the measurement of blood pressure and SPPB in the method section as follows: Line [ 234-256]

The Short Physical Performance Battery (SPPB) is a comprehensive assessment tool that measures three key physical capabilities: standing balance, walking speed, and the ability to rise from a chair. Each of these abilities is quantified through timed measurements. For the balance assessment, participants were required to assume three progressively challenging postures: side-by-side, semi tandem (heel of one foot adjacent to the big toe of the other), and tandem (heel of one foot directly in front of and touching the other foot), maintaining each position for 10 seconds. The walking speed was gauged by measuring the time it took participants to walk a 4-meter distance at their usual pace. The fastest of two attempts was recorded for the final score calculation. In the chair rise test, participants demonstrated their ability to stand from a seated position with their arms crossed over their chest. Subsequently, participants were instructed to perform five consecutive chair stands as quickly as possible, with the completion time being documented.

Resting blood pressure (BP) was assessed at the commencement of the study using oscillometer techniques, with a BP device (BPBIO320S, InBody Co, Seoul, Korea). The measurements were conducted using a blood pressure cuff that was properly fitted to the subject's dominant arm, in accordance with the current clinical guidelines [11]. On the initial visit, two blood pressure readings were taken while the participant was seated. These measurements were taken after a 30-minute period of rest. The arithmetic mean of these measurements was calculated and recorded as the BP

Q10. [198] Medication dosage is not described in Table 1. If the authors measured that, it should be clearly stated in the Materials and Methods part and the Result part.

Response 10: Thank you for highlighting this point.

It appears that we have incorrectly used the term "dosage." In this study, we did not measure the dosage but rather the number of medication intake. Therefore, we will replace "dosage" with "number of medications." Line [295]

 Additionally, in Table 1, we have indicated the proportion of participants taking four or more medications, representing polypharmacy.

Q11. [202, table1] Mean ± standard deviation can be used only when the variable is normally distributed; in the case of a non-normal distribution, it should be the median (interquartile range).

Response 11: Thank you for the suggestion. We revised the table 1.[ Line 299]

Q12. [205] One-way ANCOVA assumes a normal distribution of the population; SGDS was probably non-normally distributed, so please check carefully that your statistical methods are correct and modify them if necessary.

Response 12: Thank you for the appropriate suggestion. In figure 2, this has changed: Line [263-282, figure2]

All statistical analyses were performed using IBM SPSS V28.0 (IBM Corp., Armonk, NY, USA). Shapiro-Wilk test was used to evaluate the normality of each variable and to compare the robust and frail groups, the student t-test was per-formed when the data were normally distributed and the Mann-Whitney test when the data were non-normally distributed. To observe differences in physical function, cognitive function, and mental function, between-group comparisons between FP, FNP, NFP, and NFNP were analyzed using analysis of variance (one-way ANCOVA) when the data were normally distributed; SGDS, which was non-normally distributed, was analyzed using Kruskal-Wallis test; and Mann-Whitney U test was used for post hoc comparisons.

Q13. [207-211] The results and their interpretation are confusing. TUG, SGDS, and SMWT show differences between the groups based on the presence or absence of frailty, regardless of polypharmacy. On the other hand, for FCSST and MMSE, the effect of frailty differs between those with polypharmacy and those without polypharmacy, which indicates the combined relationship of frailty and polypharmacy.

Response 13: Thank you for the comment. We have revised the sentences as follows: Line [306-310]

Figure 2 shows that TUG, SGDS, and SMWT show differences between the groups based on the presence or absence of frailty, regardless of polypharmacy. On the other hand, for FCSST and MMSE, the effect of frailty differs between those with polypharmacy and those without polypharmacy, which indicates the combined relationship of frailty and polypharmacy. 

Q14. [221]   Please correct the two periods.

Response 14: Thank you for thoroughly reviewing our manuscript. We have corrected the double use of periods.

Q15. [226-228] According to Table 3, After adjustment, the odds ratio might be 8.66 for a shorter 6-minute walk distance and 1.97 for a lower MMSE score. Please correct the data.

Response 15: Thank you for highlighting the error in data. This was an error made while typing the numbers. We have corrected the error (Line [323-334]).  

Table 3 shows that the logistic regression model estimating the risk of frail-ty, low SMWT showed a significantly increased risk of frailty, with an odds ratio (OR) of 7.17 and a 95% confidence interval (CI) of 6.59-22.32. Low TUG showed a significantly increased risk of frailty, with an odds ratio (OR) of 2.70 and a 95% confidence interval (CI) of 1.41-5.19, Low FCSST showed a significantly increased risk of frailty, with an odds ratio (OR) of 2.36 and a 95% confidence interval (CI) of 1.22-4.58. Additionally, Low global cognitive function was associated with a 2.02-fold increase in frailty risk (95% CI: 1.06-3.83) and high depressive symptom showed a significantly increased risk of frailty, with an odds ratio (OR) of 2.10 and a 95% confidence interval (CI) of 1.05-4.22, after adjustments for age, sex, BMI, and number of medications intake, individuals with the low SMWT, low cognitive function was associated with 8.66 ( 95% CI : 4.55-16.48), 1.97 (95% CI: 1.02-3.67), respectively.

Q16. [248] Is this sentence grammatical? How about writing “In our study, we found~” or “Our study showed~”.

Previous Line [248-249] In our study, we found that frailty risk group had lower endurance capacity compared to robust group.

Response 16: Thank you for the comment. As per the suggestion, we have implemented the recommended changes as follows: Line [383]

In our study, we found that frailty risk group had lower endurance capacity compared to robust group.                

Q17. [259-261] The authors repeat the same content in two sentences, so either one is fine.

Previous Line [259-261] In our study, frail subjects had lower global cognitive function and higher depressive syndrome. The frailty risk group had lower global cognitive function compared to robust group.

Response 17: Thank you for the comment. We have deleted the duplicates and rewriting (Line [375-465]). 

Q18. [270-294] In the Discussion part, the authors’ logic is confusing and should be thoroughly reviewed. I believe the purpose and conclusion of this study is to address the issue that the current diagnostic criteria for frailty do not take into account the loss of endurance, so please discuss how to draw that conclusion. Currently, you are merely listing past studies. The first result of this study is that lower extremity endurance was strongly associated with frailty and also with a combination of frailty and polypharmacy; the second is that there was also an association between mental and cognitive function and physical function. Furthermore, in the group in which polypharmacy was combined with frailty, the effect of SGDS was different from that of frailty alone. For each of these, you should describe what actions can be taken if early recognition is made, citing proper previous studies.  

Response 18: Thank you for the comments and insightful suggestions. We have revised: Line [375-465]

Q19. [295-303] Wouldn't it be a LIMITATION to not take into account chronic diseases that do not affect exercise endurance?

Response 19: Thank you for the comment. We have incorporated the recommended suggestion as follows: Line [508-517]

First, the cross-sectional design of the study limits its ability to establish causality. Longitudinal studies are needed to clarify the temporal dynamics between frailty, polypharmacy, and functional decline. Additionally, the participant pool was limited to a specific demographic group, which may limit the generalizability of the findings. The study did not assess specific drug classes or possible drug interactions that may have different effects on frailty and functional outcomes. We also did not consider the specific comorbidities of chronic diseases that may affect endurance. Future research should include studies that examine the impact of specific medications and polypharmacy patterns on frailty and its interaction with frailty risk factors in older adults, as well as RCTs that improve upon this study.

Reviewer 2 Report

Comments and Suggestions for Authors

Title: Association between Physical Function, Mental Function, and Frailty  in Community-dwelling Older Adults: A Cross-sectional Study

Sugestion:

General Comment

This study investigates how various physical and mental functions relate to frailty, both independently and in conjunction with polypharmacy, among older adults living in the community. It significantly contributes to our understanding of the factors associated with frailty in this demographic.

Specific Comments/Suggestions

Title : Consider to revise to better describe the study: The Association of Physical and Mental Function with Frailty and Polypharmacy among Community-dwelling Older Adults: A Cross-sectional Study

Abstract

·      L12-13: Consider revising the aim statement to align with the results presented. "This study examines the relationship between physical and mental function and frailty, independently and in conjunction with polypharmacy, among older adults."

·      L20: Provide information on the classification and analysis plan for outcome/dependent variables. Recommendation: "The study examined frailty status (frail vs not frail/robust) and frailty with polypharmacy (frailty with polypharmacy vs other) using multivariate logistic regressions, adjusting for age and sex."

·      L27: Confirm and revise the statement to reflect which physical functions are associated with frailty and frailty with polypharmacy in the adjusted model.

Introduction

·      L80: Revise the aim statement to align with the analysis conducted in the study. Provide further justification for analyzing the risk of developing frailty and frailty with polypharmacy based on physical and mental functions.

Method

·      Combine sections 2.2 to 2.6 into 2.2 Outcome measures and predictor variables. Subdivide into 2.2.1 Frailty and Frailty with Polypharmacy Status (explain the binary categories of frailty status: Frail and Not Frail/Robust) and frailty with polypharmacy status (FP and other).

·      Integrate section 2.7 into this section, explaining that polypharmacy is an outcome variable dependent on frailty status. Classify frailty and polypharmacy into four classes (FP, FNP, NFP, NFNP).

·      Explain how each physical and mental function is classified into high and low categories in section 2.2.2. Outline sociodemographic variables in section 2.2.3.

·      In section 2.8, provide an analysis plan detailing comparisons between groups and the use of appropriate statistical tests.

Results

·      Report findings in the same order as the analysis plan. Include participant numbers for each category, such as in Figure 2 and Table 3.

·      Table 3:  provide participant numbers for each subgroup of physical and mental function. Include p-values in the last column.

·      In lines 224-228, report ORs resulting from the adjusted model, not just the adjusted model.

·      Apply the same comments and suggestions to Table 4 as for Table 3.

Discussion

·      Focus on interpreting findings rather than detailing numbers. Refer to the adjusted model when discussing results.

Conclusion

·      Clarify and revise statements regarding which physical and mental functions are associated with frailty and frailty with polypharmacy.

Comments on the Quality of English Language

The manuscript needs minor English editing

Author Response

Q6. [methods] Combine sections 2.2 to 2.6 into 2.2 Outcome measures and predictor variables. Subdivide into 2.2.1 Frailty and Frailty with Polypharmacy Status (explain the binary categories of frailty status: Frail and Not Frail/Robust) and frailty with polypharmacy status (FP and other).

  • Integrate section 2.7 into this section, explaining that polypharmacy is an outcome variable dependent on frailty status. Classify frailty and polypharmacy into four classes (FP, FNP, NFP, NFNP).
  • Explain how each physical and mental function is classified into high and low categories in section 2.2.2. Outline sociodemographic variables in section 2.2.3.
  • In section 2.8, provide an analysis plan detailing comparisons between groups and the use of appropriate statistical tests.

Response 6: Thank you for the comment.  We have modified the sentences in the Method:

Line [132-158],

2.2 Outcome measure and predictor variables

2.2.1 Frailty

Frailty was defined according to Linda Fried’s frailty phenotype guidelines [7]. The same method was used in our previous study [28]. Fried’s phenotype consists of five conditions as follows:

  1. Weakness- grip strength adjusted for gender and BMI.
  2. Weight loss- unintentional weight loss of 4.5 kg in the previous year.
  • Slowness of gait- calculated by the time taken to walk 4-meter twice without assistance.
  1. Exhaustion- self-reported exhaustion
  2. Low-dose physical activity- calculated based on accelerometer data as per the recommendation for physical activity in older adults by ACSM.

The frailty risk group was defined according to Linda Fried's criteria as in-cluding both the frail and prefrail groups. In other words, participants who met any one of the previously mentioned criteria were categorized into the frailty risk group, while those who did not meet any criteria were considered part of the ro-bust group.2.2.2 Frailty with polypharmacy

Polypharmacy was defined as intake of more than 4 drugs [23]. The number of drugs usually consumed was determined from the participant’s prescriptions. Medication used on a regular basis, excluding vitamins and mineral supplements, was recorded. The participants were further categorized into four groups based on presence and absence of frailty, polypharmacy and/or both as follows:

  1. Frail with presence of polypharmacy (FP)
  2. Frail and absence of polypharmacy (FNP)
  • Robust with presence of polypharmacy (NFP)
  1. Robust and absence of polypharmacy (NFNP)

Q6. [results] Report findings in the same order as the analysis plan. Include participant numbers for each category, such as in Figure 2 and Table 3.

  • Table 3: provide participant numbers for each subgroup of physical and mental function. Include p-values in the last column.
  • In lines 224-228, report ORs resulting from the adjusted model, not just the adjusted model.
  • Apply the same comments and suggestions to Table 4 as for Table 3.

Response 6: Thank you for the comment. We have revised.

Line [344,366- table3, table4]

Q7. [discussion] Focus on interpreting findings rather than detailing numbers. Refer to the adjusted model when discussing results.

Response 7: Thank you for the insightful comment. We have revised the discussion as your comment.

(Line [374-465]):

Q8. [Conclusion] Clarify and revise statements regarding which physical and mental functions are associated with frailty and frailty with polypharmacy.

Response 8: Thank you for the comment. We added the measurement as follows: Line [530-531]

In addition, the measures did not consider chronic diseases that may affect endurance.

Round 2

Reviewer 1 Report

Comments and Suggestions for Authors

The manuscript was thoroughly reviewed and most of the points raised were improved. Please reconfirm the following points.

1.     Line 101 Figure 1

Since the inclusion criteria are 60 years and older, you do not need to list being under 60 years as exclusion criteria.

2.     Line 245 Table 1

In response to Q11, the authors have corrected the statistical method for non-normal distribution (living alone, SF-GDS), but the table legend has not been corrected.

3.     Line 249-, Figure 2

In response to Q12, the authors have corrected the description of the statistical method in line 224-228, however, the description in the result part and the figure legend were not corrected. Please describe exactly which statistics you have chosen.

4.     Line 275

It also seems to have double periods.

Author Response

Response to Reviewer’s comments / Manuscript ID: jcm-2962564

Reviewer #1

Q1. [Line 101 Figure 1] Since the inclusion criteria are 60 years and older, you do not need to list being under 60 years as exclusion criteria.

Response1: Thank you for your comments. I have changed Figure1. Line [101, figure 1]

Q2. [Line 245, Table 1] In response to Q11, the authors have corrected the statistical method for non-normal distribution (living alone, SF-GDS), but the table legend has not been corrected.

Response2: Thank you for your comments. I added in Table 1. Line [258-260]

Data are represented as mean ± standard deviation, median (interquartile) and n (%); Variables were compared by T-test, a Chi-square test, bMann–Whitney test, * p<0.05.

Q3. [Line 249-, Figure 2] In response to Q12, the authors have corrected the description of the statistical method in line 224-228, however, the description in the result part and the figure legend were not corrected. Please describe exactly which statistics you have chosen.

Response 3: Thank you for the appropriate suggestion. we modified Results section: Line [230-240, 276-277]       

Line [230-240] Descriptive statistics for continuous variables were delineated as means ± standard deviations for distributions adhering to normality, and as medians complemented by interquartile ranges for non-normally distributed data. Categorical variables were quantified as frequencies and percentages. The comparative analysis between robust and frail groups involved the application of Student's t-test for continuous variables with normal distributions and the Mann-Whitney U test for those with non-normal distributions. The Chi-square test was utilized to examine associations among categorical variables. Differences in physical, cognitive, and mental functions across groups were assessed using one-way Analysis of Covariance (ANCOVA) for normally distributed continuous variables, and the Kruskal-Wallis test for variables not following a normal distribution.

Line [276-277] One-way analysis of covariance (ANCOVA) with the Bonferroni test and a Kruskal-Wallis test with the Mann-Whitney U test were used compare frail & polypharmacy (FP), frail & non-polypharmacy (FNO), robust & polypharmacy (NFP), and robust & non-polypharmacy (NFNP); The error bars represent the standard error of the mean. *p<0.05, **p<0.001

Q4. [Line 275] It also seems to have double periods.

Response 4: Thank you for thoroughly reviewing our manuscript. We have corrected the double use of periods.
